# Retrospective Analyses of COVID-19 and Population Ageing Effects on Italian Mortality During the Pandemic

**DOI:** 10.3390/ijerph20156481

**Published:** 2023-07-31

**Authors:** Damiano Brunori, Giovanni Vanni Frajese, Emma Sarno

**Affiliations:** 1Department of Computer, Control and Management Engineering, Sapienza University of Rome, Via Ariosto 25, 00185 Rome, Italy; 2Department of Sports Science, Human and Health, University of Rome ‘Foro Italico’, Piazza Lauro de Bosis, 15, 00135 Rome, Italy; giovanni.frajese@uniroma4.it; 3Department of Human and Social Sciences, University of Naples L’Orientale, Largo San Giovanni Maggiore 30, 80134 Naples, Italy; esarno@unior.it

**Keywords:** COVID-19, retrospective analyses, mortality rate, age groups, population structure, Italian pandemic, ageing

## Abstract

The spread of COVID-19 led to an extremely high number of deaths in Italy in 2020 with respect to previous years. Because the total number of deaths may depend on both the population structure and the mortality rate by age groups, a detailed overview of the Italian pandemic situation is here provided by following two main lines of inquiry: (i) checking for similarities and differences among mortality rates per age groups before and during the COVID-19 spread; (ii) analyzing the responsiveness of the Italian population structure to different mortality rates. Real-based evidence led us to conduct analyses for two groups associated with different population stages of life, referred to as younghood and adulthood periods. We focus on the Italian pandemic from February 2020 to March 2021. Our study helps to understand why elders dramatically impacted the total number of deaths. In addition, it reveals how badly the 2020 Italian population structure would have reacted to mortality rates already faced in the past. Finally, politicians, scientists, and journalists’ statements and other ways of communicating information about COVID-19 are questioned in the light of scientific data available at that time.

## 1. Introduction

A detailed diachronic reconstruction of the events that occurred in Italy during the first COVID-19 pandemic period is needed to put our findings into proper perspective. The first diagnosed case of SARS-CoV-2 among the Italian population was on 21 February 2020 [1], even though a retrospective analysis discovered SARS-CoV-2 RNA gene sequences through the biopsy of a young Italian woman in November 2019 [2]. During the period 2020–2021, the ISS (Italian National Institute of Health) produced several reports describing the characteristic of patients who died positive for SARS-CoV-2 infection, whilst ISTAT (Italian National Institute of Statistics) provided more detailed data about the overall Italian population mortality rates (MRs) by reporting the causes of death by age groups; however, in the latter case, data were not always promptly updated and available on the date on which they were consulted (as explained in Section 3). Notice also that the beginning of the anti-COVID-19 vaccination campaign in Europe started on 27 December 2020 [3] (see Figure 1 for better temporal framing of the occurred events).

During 2020–2022, most politicians, scientists, and journalists disseminated messages of maximum alert addressing the entire Italian population concerning COVID-19. The mass media constantly invited Italian people to exercise maximum caution, substantiating the alert with daily updating about the number of deaths due to COVID-19 (e.g., [4]). Some journals referred to the death of younger people caused by COVID-19, such as the case of a young Italian cyclist [5] or that of a two years old child "under desperate conditions" due to COVID-19 [6], while many other journals daily marked the severity of the situation. In this context, the entire Italian population undoubtedly experienced tension, fear, and uncertainty regarding the pandemic.

In the scientific literature, retrospective studies on the Italian pandemic situation focus on strategies to stem the spread of COVID-19, also evaluating the government intervention policy [7], while others reconstruct the epidemic wave with all their factors to help in better understanding of how to face this kind of exceptional event [8]. Another work [9] deals with the impact of COVID-19 on the Italian population mortality, analyzing the number of all the causes of death for both females and males and highlighting the region where COVID-19 caused a higher number of deaths. Our retrospective analyses instead highlight how the pandemic differently affected the Italian mortality of distinct ages in 2020 and how the daily official reported news about many age groups was poorly consistent with what was actually happening. The investigated period ranges from February 2020 to March 2021, namely when vaccines were unavailable or very limited. In order to avoid the risk of generalizing concepts and opinions that actually differ enormously from case to case, our work plumbs the Italian population structure along with its MRs to offer a deeper insight into the mechanism that led to the well-known excess of deaths during the COVID-19 emergency in Italy. Our main goal is then to point out the main differences that arise in terms of total deaths with respect to the past, aiming at assessing for which age groups and to what extent the number of deceased people might be considered exceptional. The total number of deaths recorded each year can be considered as the combined result of two separate factors, i.e., population ageing and the mortality rate. Hence, we explore the actual role of these two factors during the pandemic by experiencing what would have happened if one of them had been fixed in the past while the other had changed. In order to avoid a misleading and inaccurate interpretation that may arise from aggregated values related to the entire population, we divide the inhabitants into equal size classes of five years. This choice also allows us to perform a better comparative analysis of the causes of death for young people. In addition, similarly to Heligman and Pollard [10], who decomposed the MR pattern according to specific characteristics of different age groups, we decide to split the population into two groups corresponding to the stages of life in the individuals, namely *younghood* (including people aged 0–49 years old) and *adulthood* (including people aged 50–95+ years old). This population subdivision highlights that the number of people in the younghood category has been decreasing since 2011 (from 36,043,544 to 32,474,853 inhabitants in 2020), whereas the number of people in the adulthood class has been increasing since 2011 (from 23,904,953 to 27,166,635 inhabitants in 2020).

The rest of this paper is structured as follows. Section 2 describes in detail the results of the main two investigations carried out on: (i) the mortality rates per age groups along with the population structures referred to analogous classes; (ii) the Italian population’s responsiveness to the spread of COVID-19. Section 3 contains some considerations on how the results can be interpreted by also providing some limitations of this investigation and suggesting a few future work directions, and Section 4 contains a summary of this work, along with some final remarks.

## 2. Retrospective Analyses

The two following parallel analyses are presented in this section:*Italian MRs and population structure by age groups*. Since the impact of COVID-19 was extremely different in terms of deaths among younger and elderly people, we mainly focus on adulthood (i.e., where the pandemic hit hard), trying to figure out how hard it was compared to what happened during previous years 2011–2019. Thus, we examine the effect of the ageing population through the variations that occurred in both the population percentages and the MRs for the same age groups and in the same period. For what concerns younger people, whose mortality did not increase during the pandemic, we contextualize COVID-19 mortality in a broader framework where other causes of death are considered.*Italian population responsiveness*. Two parallel simulations are carried out to assess the responsiveness of the Italian population to different MRs.

### 2.1. Italian MRs and Population Structure by Age Groups

Just looking at the total mortality rate values could blur what happens inside specific age groups, leading to an incorrect generalization or interpretation of the data. Indeed, even though Table 1 shows an increased mortality rate in 2020 associated with the entire population, it does not reflect the trend of most of the age groups composing it. It is also worth noticing an overall increasing trend in the Italian MR from 2011 to 2021.

Therefore, we investigate the mortality rate trend by age groups using ISTAT data [11,12], which can be easily organized in classes of five year spans. In particular, limiting our attention to people under 40 years old in Italy, we observe how that the MRs trend has been decreasing since 2011, as represented by the linear interpolation given by the blue downward-sloping line in Figure 2: this trend is definitely in contrast with what is shown in Table 1.

No variation exists in the mortality rate between 2020 and 2019, while 2021 shows very little but no significant increase in the mortality rate (indeed, it is still lower than the MR values associated with the years before 2019). Thus, it is quite straightforward that COVID-19 did not impact the mortality of the younger population in Italy, as also shown in Figure 3, where the mortality variation per thousand inhabitants aged 0–64 years old between 2019 and 2020 is represented. Therefore, only with these data, we may certainly conclude that in 2020, COVID-19 had:Zero impact on mortality for people aged 0–4, 5–9, 10–14, 15–19, 25–29, 20–34, and 40–44 years old (notice also that the COVID-19 lethality for people aged 0–39 years old is so low that in March 2021, ISS reported a value equal to 0 associated with it [13]);A negligible impact (−0.1 mortality variation, i.e., fewer deaths, in 2019–2020) on the mortality for people aged 20–24 years old;An irrelevant impact (+0.1 mortality variation, i.e., more deaths, in 2019–2020) on the mortality for people aged 35–39 and 45–49 years old;An increasing impact on mortality for people aged 50+ years old, but still not significant for those aged 50–54 years old and 55–59 years old (respectively, +0.2 and +0.3 mortality variation in 2019–2020).

Nevertheless, we included the people aged 50+ years old in the adulthood category, as it is well-known that this age group was considered the most critical during the pandemic. The ISS report on 5 March 2021 [14] confirms that the younger population mortality was not affected by COVID-19 (also note that the number of infected people in Italy was higher than the one detected [15], and thus the COVID-19 lethality would have been even lower than the one reported at that time). For scale reasons, the mortality rate for people aged 65+ years old is not reported in Figure 3. Indeed, because of the high variation between 2020 and 2019 in the MR for people aged 70+ years old, it would not have been possible to represent the tiny variations (where present) for young people on the same graph.

For the sake of completeness, we report the mortality rate variations (per thousand inhabitants) among people aged 65+ years old between 2019 and 2020: (i) +1.5 for 65–69 years old; (ii) +2.9 for 70–74 years old; (iii) +4.9 for 75–79 years old; (iv) +8.5 for 80–84 years old; (v) +14.7 for 85–89 years old; (vi) +27.9 for 90–94 years old; (vii) +43.49 for 95+ years old. Thus, a significant and negative impact (i.e., more deaths) of COVID-19 on mortality could possibly be related only to 80+ years old age groups. No further analysis in this regard is needed for people under 50 years old, as their mortality rates in 2011–2019 either did not change or were completely negligible (Figure 2): analogous reasoning can be done for their mortality rates related to 2020 with respect to 2019 (Figure 3).

In the next sections, the adulthood (i.e., the age period referring to 50+ year old people) will be examined in more detail.

#### 2.1.1. Adulthood

Our scope is to assess whether the 2020 MRs for all the adulthood age groups can be considered consistent or not with the MRs related to the same age groups in the years ranging from 2011 to 2019. In order to achieve this aim, we can benefit from two different and complementary analyses: (i) detection of outliers in 2020 MRs with respect to the past years; (ii) usage of distance matrices for evaluating the time similarity between pairs of data distributions related to MRs and population shares in the 50+ year old people.

Table 2 provides a finer comprehension of the mortality trends for people aged 50+ years old, thus allowing us to address point (i). The table reports different parameters associated with the mortality rates in 2011–2019 by comparing them with those present in 2020. The data referred to in the period 2011–2019 are scarce, and thus their distribution cannot be inferred properly: however, they are sketched in Section A.3 (some distribution results are more skewed than others). Hence, aiming to check for outliers, the Z-score and the IQRs-method are used. It is well known that a Z-score greater than |3| indicates that the corresponding value can be an outlier (under the hypothesis of a normal distribution) with a probability of occurring not greater than 1%. However, since the dataset available for this analysis is made up of nine observations, we will consider |2| as the absolute Z-score threshold to detect values with a probability to occur not greater than 5%. In Table 2, Z-scores are associated with the 2020 MR value and computed with respect to the mean and standard deviation of the MRs computed over the period 2011–2019. Furthermore, alternative robust non-parametric intervals not depending on parametric distribution assumptions are provided. Indeed, applying the IQRs-method, we consider as extremes those equal to $\left(Q_{1}-1.5IQR,Q_{3}+1.5IQR\right)$, where $Q_{1}$ and $Q_{3}$ are the first and the third quartile, respectively. Table 2 also includes the range (i.e., minimum and maximum detected values) and the median referred to MRs in 2011–2019. As a result, we can observe a larger value than the upper bound of the MR’s range only for 70–79 and 85–95 year old age groups; a similar outcome can be observed for Z-scores greater than 2. The upper bound of the 2011–2019 IQR is exceeded only by the 2020 MR of the 90–94 year old age group. Notice that for age groups of 50–64 years old, the 2020 mortality rate is the same as the mean and the median values related to the period 2011–2019. All these things considered, we can reasonably infer that only a few age groups affected the mortality rate increase associated with the whole population in 2020.

A further analysis is now carried out by examining the entire set of values corresponding to each age group present in adulthood, year by year. In particular, to figure out how each annual MR distribution by age groups in adulthood could be considered similar (or dissimilar) compared to past ones, we build up two distance matrices based on the Euclidean and the maximum distances, respectively. Results are reported in the reddish triangular lower part of the matrices in Figure 4, where the Euclidean and the maximum distances between the inhabitants’ percentages belonging to the same age groups for each pair of years are also included in the bluish upper triangular part of such matrices. During the decade 2011–2021, the Italian population suffered from constant and progressive ageing, and thus the examination of all the variations that occurred in the population structure is essential to control such an ageing effect. Details on the population ageing are shown in Section A.2 (Figure A1) through a composite bars chart, where, for the sake of simplicity, only four age categories are considered. Notice also that the population ageing is further strengthened by the increase in the absolute number of inhabitants in adulthood.

The distances between each pair of years (t−1,t) allow us to understand which distribution behaved most similarly (or dissimilarly) over two consecutive years, i.e., t−1 and *t*. In particular, small distance values indicate that no abnormal events may have occurred during the transition from one year to the next, whereas big distance values indicate the opposite. Similarly, the years’ pairs (t−k,t), provide information about what happened between two non-consecutive years distant from each other by a given lag *k*. The evolution of the population structure associated with the ageing process appears rather smooth in time, as all the distances are quite similar at a given lag *k* (this is highlighted by a uniformly stronger blue colouring at increasing lags in the upper triangular part of the matrices in Figure 4). Nevertheless, the biggest variation in the 50+ year old population between two consecutive years corresponds exactly to the transition from 2019 to 2020 (i.e., in correspondence with the COVID-19 spread), followed by those that occurred in (2015,2016) and (2014,2015). Note that in the period (2015,2016), a relevant flu season occurred associated with a possible harvesting effect on the population [16]. In the transition from 2019 to 2020, the age classes that contributed most to the increase in the Euclidean distance value of the 50+ year old population, are (in order) those related to 55–59, 70–74, and 60–64 years old. This means that the exceptional mortality rate equal to 18.3 for 70–74 year old people, previously seen in Table 2, hits right where an increase in the number of individuals due to ageing occurred within the same age group. A fatal condition occurred whereby COVID-19, mainly dangerous for the elderly, hits a population where the share of the elderly people had increased more than others. Moving on to MRs distances, we notice that variations occur quite irregularly over time. However, it is possible to observe that the MR distance in the years’ pair (2019,2020) is definitively the biggest over the analyzed decade, whatever the type of distance considered, namely either Euclidean (Figure 4a) or maximum (Figure 4b). In addition, abnormal transitions had occurred similarly in the past, although toned down, in (2014,2015) and (2015,2016), i.e., when a severe flu season hit Italy. The analogy between the mortality rates in 2020 and 2015 is likewise evident by noticing that the minimum value of the distances concerning the 2020 MR distribution corresponds exactly to 2015; conversely, the 2020 percentage of the 50+ year old population is quite distant from that present in 2015.

#### 2.1.2. Younghood

This section compares the risk of younger people dying from COVID-19 with respect to other diseases. We will focus on people aged 0–39 years old, which is the only category associated with the ISS reports [17,18] containing every detail related to all the patients deceased due to COVID-19. For information concerning the first causes of death in young people, we refer to the ISTAT data [12]. Since we know that the 2020 MRs related to the people aged 40–49 years old are perfectly consistent with the past ones (see Figure 3), we are not excluding relevant information from our discussion. Following the ISS reports [17,18], patients counted in the total of deaths caused by SARS-CoV-2 are all the “deceased SARS-CoV-2 patients positive”. Since most patients are associated with “serious pre-existing pathologies (cardiovascular, renal, psychiatric disorders, diabetes, obesity)”, two main classes are defined here to classify COVID-19 deaths. One class (COVID-19 O) is related to the overall deceased SARS-CoV-2 patients positive, including both the patients with and without serious pre-existing diseases, and another one (COVID-19 NPP) only refers to deceased SARS-CoV-2 patients positive without any serious pre-existing pathology. The latter class represents a subset of the former. Note that the COVID-19 O class also includes all those deceased SARS-CoV-2 patients positive “with no clinic info available”: these patients’ cases were equal to 41 and 62 on 16 December 2020, and on 1 March 2021, respectively. Therefore, the COVID-19 O category allows us to look at the worst possible case, even without specific patient information. As for periodization, two different dates are used in our analyses to close the pandemic interval that started on 21 February 2020, and both dates are chosen to correspond with two released ISS reports [17,18]. In more detail, the period from 21 February 2020 to 21 December 2020, identifies the UV period (i.e., period of unavailable vaccines) covering about ten months of deaths, possibly due to COVID-19 and in which anti-COVID-19 vaccines were unavailable. The period instead from 21 February 2020 to 1 March 2021 covers the first year of the COVID-19 spread in Italy without the availability of anti-COVID-19 vaccines for the vast majority of people (mainly the younger ones). Indeed, the COVID-19 vaccines could be booked based on scheduling, prioritizing the most vulnerable people or the elderly (over 80 years old) or those working in health or similar facilities [3]. Hence, we will refer to this time interval as the LOUV period (i.e., period of very limited or unavailable vaccines): this twelve-month period is necessary for comparing ISS data with ISTAT annual data, as the latter are related to the calendar years. Even if the ISS period (LOUV) begins approximately three months after the ISTAT period (calendar year), they both cover twelve months and hence they can still be fairly compared with each other in terms of events’ frequency. To better contextualize our inspection, we provide further details below about the pandemic situation in force during the considered period. The so-called *Vaccine Day* (27 December 2020) [3] marked the beginning of the anti-COVID-19 vaccination campaign throughout Europe (and thus also in Italy). The Italian national update on the COVID-19 epidemic situation provided by ISS on 3 March 2021 [19] stated that the overall number of administrated doses of the anti-COVID-19 vaccines was 4,784,690, of which 3,285,421 were first doses and 1,499,269 were second doses. In addition, we know that the number of doses administrated to people aged 80+ years old was 809,698. Thus, the number of anti-COVID-19 vaccine doses administrated to the population in the age group 12–80 years old was 3,974,992: children 0–12 years old were not receiving any doses at that time. Since we can neither compute the number of vaccine doses administrated to the people aged 0–39 years old with sensible precision nor exactly know it for lack of data availability, we will try to estimate it naively. Assuming that the doses were distributed equally among the people aged 12–80 years old and knowing the number of resident inhabitants in 2020 [20], we can estimate the percentage of the population in the age group 0–39 years old who received at least one vaccine dose, which can then be equal to 6.85%. Note that this is quite a substantial overestimate given that the vaccine scheduling procedure prioritized people aged 40+ years old with respect to those aged 0–39 years old, and by also knowing that people in 13–22 years old could not work in a prioritized sector (e.g., health), obviously mainly due to the minimum age needed for the title. Thus, it should be noted that the percentage of vaccinated people in the 0–39 year old age group on 3 March 2021 was, reasonably, far lower than 6.85%. In addition, we also need to consider that it refers to the number of doses administrated and not to the fully vaccinated condition, i.e., two vaccine doses at that time. For the sake of completeness, it must be highlighted that the date of the vaccination report performed by the ISS refers to 3 March 2021, while the date associated with the COVID-19 deaths’ report (still by ISS) refers to 1 March 2021, and this is because the two reports were made on different days. The difference of two days is obviously not relevant to this investigation.

All these things considered, we can compare the number of deaths due to COVID-19 in a year (LOUV period) with deaths caused by completely *external and unpredictable events*. Figure 5 shows that the number of deceased patients for which COVID-19 is the first cause of death (in both UV and LOUV periods) is significantly lower than the ones for which the first cause of death is represented by suicides, murders, and accidents in 2020.

It is worth noticing that the number of deaths due to COVID-19 for people aged 0–39 years old in only two and a half months (i.e., from 16 December 2020 to 1 March 2021) increased by 34% and a 90% in COVID-19 O and COVID-19 NPP, respectively. This could likely be explained by the fact that the year for counting COVID-19 deaths ranges from 21 February 2020 to 2 March 2021 and could include an extra peak of the epidemic “winter’s wave”. Thus, in a regular calendar year, namely, from the beginning of January to the end of December, we should expect fewer deaths than those reported in the LOUV period: however, using upper bounds avoids the risk of underestimation.

The summary Table 3 shows the ratios between the cases (among young people, i.e., 0–39 years old, in Italy) reporting a first cause of death different from SARS-CoV-2 and those associated with it in 2020 and in the LOUV period, respectively.

Thus, the values reported in this table indicate how many times the considered *Non-COVID-19 illness (event)* occurred more or less than the *COVID-19 first causes of death*: a value greater than 1.0 shows a higher frequency for *Non-COVID-19 illness (event)*, whereas a value lower than 1.0 shows a higher frequency for *COVID-19 first causes of death*.

The causes of death different from COVID-19 are selected based on two main factors: (i) relationship with external causes due to unpredictable events; (ii) association with some malformations or tumours. Mostly, causes of death included in (i) and (ii) are joined by not being really preventable (e.g., suicides, accidents, malformations) and thus less overlappable with other diseases that may lead to death.

It is evident that all the people aged 0–39 years old (with and without pre-existing pathologies) in the LOUV period were still more likely to die because of external and unpredictable events than COVID-19 (Table 3). Indeed, for example, for young people without pre-existing pathologies, even the causes of death due to murders and aggression were more frequent than those due to COVID-19.

### 2.2. Italian Population Responsiveness

The impact of COVID-19 on populations characterized by different age structures is investigated in this section through two complementary analyses. Similarly to our previous analyses, we consider age groups of five year spans (e.g., 20–24 years old, 55–59 years old) across the entire population in order to assess either the effect of different MRs on invariant populations or the effect of different populations’ shares associated with each age group (the latter under the hypothesis of constant MRs). In brief, we aim to answer these two main questions:Question1: What would have happened if COVID-19 had spread before 2020 in Italy, hitting an overall younger population?Question2: What would have happened in Italy if the past populations structured as in 2020 had faced the past yearly MRs?

#### 2.2.1. Question 1: Past Populations Facing the 2020 MRs

Here, we check for a hypothetical 2020 MR impact on past Italian population structures. It is assumed that the 2020 MR in Italy was mostly affected by COVID-19 [9] and all the restrictions associated with it (e.g., limited social life, heavily restricted access to routine hospital medical checks), and therefore determining an unexpected excess of deaths. We tackle this issue through a simulation that keeps fixed the 2020 Italian MRs for each age group while making the population structure systematically vary according to the configuration before 2020 (refer to the ISTAT database and its summarizing table for resident inhabitants and reconstructed population [20,21,22]). The outcomes tell us how many deaths there would have been if COVID-19 had appeared in the past, hitting previous population structures.

Figure 6 shows the actual number of deaths in 2002–2020 (red line) and the simulated number of deaths obtained by keeping fixed the 2020 MRs by age groups and varying the population structure based on its configuration at the reference year (blue line).

It is worth noticing that the difference between actual and simulated deaths is negligible in some different years (e.g., 2004, 2005, 2011), whereas in other ones, i.e., 2002 and 2003, there would have been around 20,947 and 40,525 fewer deaths, respectively, than occurred. These findings seem inconsistent with the assumption that most of the exceeding mortality is attributable to COVID-19. Conversely, the population structure turned out to be a crucial factor in the increased number of deaths in 2020. Figure 6 shows how COVID-19 would have had no impact if it had spread throughout an Italian population structured as some of those present in the past: the closer the blue line is to the red line, the more the impact of COVID-19 on the considered population would have been negligible. Strikingly, the blue line goes under the red line for 2002 and 2003 years, meaning that the COVID-19 impact would have caused even fewer deaths than those that occurred.

#### 2.2.2. Question 2: Past MRs Applied to Populations Structured as in 2020

We check for the 2020 Italian population responsiveness to past MRs. The second question can be answered with a simulation that systematically applies the past MRs by age groups to a past population restructured as in 2020 (basically, the total number of inhabitants in the years before 2020 is rearranged into the well-known age groups using the 2020 shares). Similarly to the case of the first question, the assumption here is that COVID-19 created the emergency reflected in the higher (and “abnormal”) 2020 MRs by age. Differently, past MRs cannot be considered “abnormal” in both statistical terms and political/health measures, and, in fact, they were treated as such. The period of interest is 2011–2020 since MRs per age group before 2011 are not available on the ISTAT website. Figure 7 shows the actual number of deaths in 2011–2020 (red line) and those simulated (blue line).

Also in this case, we can see how population age largely affects the number of deaths. Indeed, the total number of deaths would have been significantly larger than those that occurred in the past years if the percentage distribution of the resident population by individual age group had been the same as in 2020: for addiction, the number of simulated deaths for some years (e.g., 2011, 2012, 2015) would have been not very far from that of 2020. This evidence shows that population ageing is a big issue in terms of expected deaths. Elderly people are more fragile and exposed to medical issues and, unfortunately, the Italian population is getting older and older: indeed, the percentage of people aged 50+ years old has been following an overall increasing trend since 2011 (see Figure A1 in Section A.1).

## 3. Discussion

Looking at the results obtained from the data analyzed so far, we can discuss and interpret them with respect to the Italian pandemic situation.

It is clear that the Italian population’s age significantly affected the mortality rate, whereas COVID-19 led to increased deaths almost entirely due to some specific age groups. The 2020 MR associated with people aged 0–39 years old was not affected at all, and only those aged 70–79 years old and 85–94 years old presented an anomalous value of the 2020 MRs with respect to those that occurred in 2011–2019. In addition, we noticed that people aged 0–39 years old were more likely to die from many other (and often external and unpredictable) causes than from COVID-19: this evidence is even more marked when people do not present any pre-existing pathology. The responsiveness of the population (Section 2.2) could have been tested to see its reactivity to past MRs and realize that it would not have faced most of them very well in terms of expected deaths. Hence, it should have been possible and necessary to understand that an ageing population might have struggled in facing some particular virulent flu periods such as, for example, that reported by ISS in 2015 [24] (hence the relevance of constantly updated pandemic plans). Furthermore, even though the evaluation process related to the ISTAT data on the causes of death takes around three years [25], in 2020–2021, it was still possible to get access to the order of magnitude of the non-COVID-19 causes of death since 2011 in people aged 0–39 years old: this might have helped to properly quantify the severity of COVID-19, avoiding generating irrational fear behaviours (See Figure A2 in Section A.2).

### 3.1. Limitations and Future Work

Some limitations of this investigation need to be considered. The analysis associated with the MRs for each age group (Table 2) is based on the accessible data provided by the ISTAT database. We provide a detailed overview of the increase of the 2020 MR in Italy, performing different but complementary analyses, focusing on the MR of the elderly and younger people and on the responsiveness of the 2020 Italian population per age group. This work cannot catch all the hidden factors that could have affected the increase of the aggregate Italian mortality rate in 2020. Some already investigated factors are, for example, related to gender [26], pollution [27], and latitude: a very high mortality rate has been identified in the northern Italian regions [28].

Further aspects that deserve to be investigated in the future for a better understanding of the factors that led to an excessive number of deaths in 2020 can be represented by the increase in the number of the first causes of death due to “*symptoms, signs, abnormal results, and ill-defined causes*” and “*unknown and unspecified causes*”. Indeed, according to the ISTAT data [12] related to the first causes of death in 2020, these values are equal to 9569 and 15,128, respectively, and are undoubtedly outliers (see Table 4).

Another controversial aspect associated with some unclear first causes of death is represented by 4742 cases reported by ISTAT under the “*COVID-19, virus not detected*” heading [12]. Summing the part of the 2020 causes of death in Table 4 exceeding their corresponding IQR’s upper bound in 2011–2019 with the number of COVID-19 deaths not associated with virus detection, we get 16,769 potential abnormal deaths due to unclear causes in 2020. This finding and the already discussed Italian population ageing undermine the common assumption that the increased 2020 MR is largely and straightforwardly due to the spread of COVID-19.

Future investigation to figure out what happened in other countries worldwide in terms of population responsiveness could also favour the contextualization of the Italian pandemic situation.

### 3.2. General Considerations

Although no comprehensive and systematic analysis of all the claims made by journalists, politicians, and scientists is carried out here, we detected a few contradictions between the claims from prominent people with important roles in the pandemic management and the data analyzed in this work. A non-exhaustive list of representative examples of these statements will follow. We can mention the headline of a newspaper article on 2 December 2021 [30], reporting the words of Dr. Brusaferro (i.e., the ISS president): “*the most affected age group is between 20 and 30 years old. Cases among children are rising*”. This kind of headline can be misleading without a proper further reading of the content, as it does not immediately specify whether “*most affected*” refers to the number of deaths or the number of infected people. A different newspaper headline [31] in 2021 focused on COVID-19 deaths in young people, referring to a specific case of a 14 year old boy who died due to COVID-19. Furthermore, in early 2022, another headline still addressed the attention on the COVID-19 situation associated with young people, reporting the words of Alessio D’amato, the Health Councilor of the Lazio Region: “*Virus also hits young people hard*” [32]. It has been noticed and marked how the number of deaths caused by COVID-19 is affected by specific age groups and how low young people’s chances of dying from COVID-19 are compared to other diseases. Thence, specific journal articles focusing entirely on very limited and rare death cases contributed to a misleading conception of the pandemic situation. It is also worth noting that Mario Draghi, i.e., the Italian Prime Minister in 2021–2022, during the press conference on 22 July 2021 [33], stated as follows: “*You don’t get vaccinated, you get sick, you die. Or kill*”. The latter is undoubtedly one of the most iconic miscommunications of scientific content of the last years: it highlights even more how the information provided by media and public figures (to protect public health) has actually relied on a generalized and imprecise view that did not comply with the scientific data available at that time. As extensively shown, COVID-19 is a risky disease only for some age groups and is largely associated with specific pre-existing clinical conditions. Thus, spreading a generalized message not reflecting all the different and equally relevant details about the real data on COVID-19 mortality should be avoided.

## 4. Conclusions

We divided the Italian population into two main age classes: Younghood (0–49 year old people) and Adulthood (50+ year old people). Then, we carried out two different analyses through the data provided by ISTAT and ISS, namely the most authoritative and accredited Italian institutes regarding information about inhabitants and their diseases. The evidence discussed here confirms that COVID-19 did not impact the mortality rate of young people in Italy in 2020 and early 2021, i.e., when COVID-19 vaccines were unavailable or very limited. Significant mortality increase with respect to the years 2011–2019 can be noticed only for age groups 70–79 years old and 85–94 years old, whereas the increase of the aggregated mortality rate is not only correlated to the spread of COVID-19 but seems to be also related to the Italian population ageing, which is well-represented by the increase of the inhabitants associated with the 50+ year old age group. Two matrices are built to show the distribution distances between the MRs and the population’s shares by age groups in 50+ year old people during the decade 2011–2020, allowing us to better understand what happened in Italy in 2020 with respect to the most recent past. Furthermore, the Italian 2020 population reaction to mortality rates (by age) that occurred in the past shows how poorly resistant it would have also been to pre-COVID-19 conditions. Finally, we point out that young people in Italy were still more likely to die due to a completely external and unpredictable event (e.g., accidents, suicides) than COVID-19. All these things considered, it is evident that most of the imprecise claims about a generalized high mortality rate due to COVID-19 perpetrated through the mass media in 2020–2021 by many influential people such as politicians, scientists, journalists, often were based on alarmist and generic messages rather than on real scientific foundation.

## Figures and Tables

**Figure 1 ijerph-20-06481-f001:**
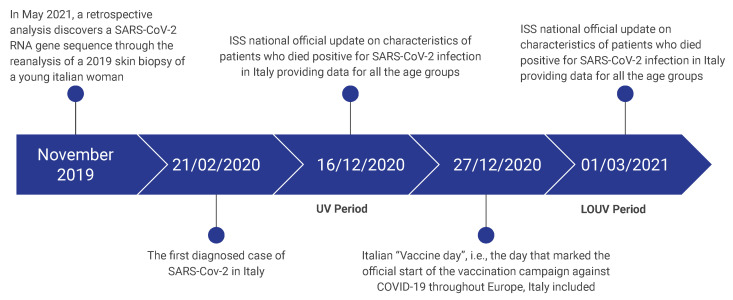
The timeline of the main events that are considered in this paper. UV and LOUV stand for Unavailable Vaccines period and Limited Or Unavailable Vaccines period, respectively.

**Figure 2 ijerph-20-06481-f002:**
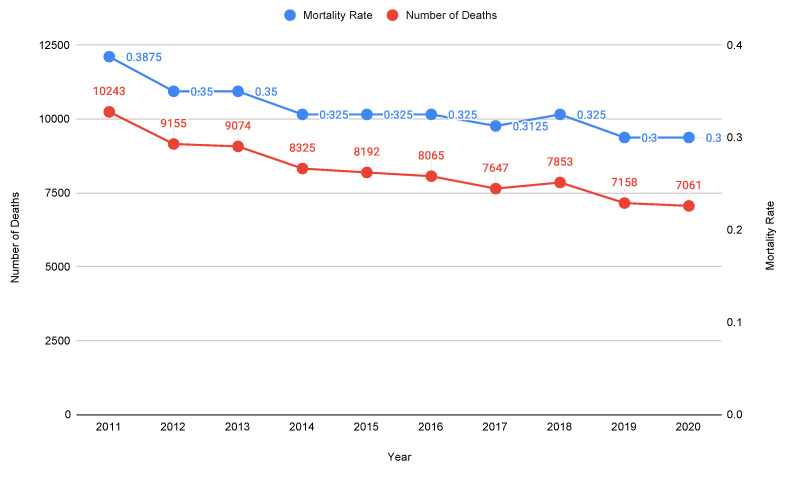
MRs (per thousand) for people aged 0–39 years old in Italy: the number of deaths computed based on the MRs reported is also represented. Authors’ elaboration on ISTAT mortality tables [11].

**Figure 3 ijerph-20-06481-f003:**
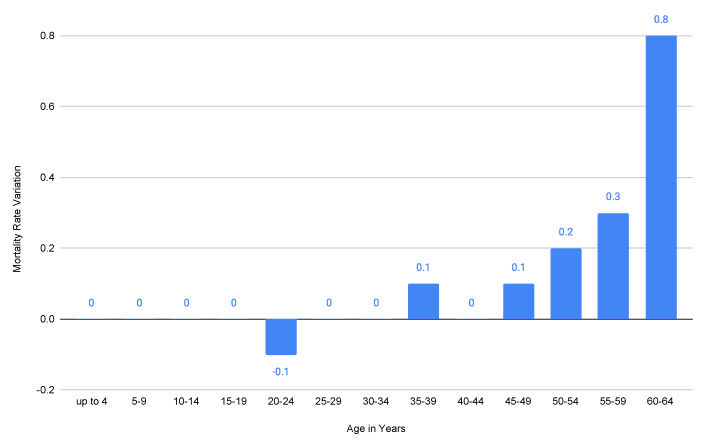
MRs (per thousand) variation between 2020 and 2019 in 0–64 year old Italian population. Authors’ elaboration on ISTAT mortality tables [11].

**Figure 4 ijerph-20-06481-f004:**
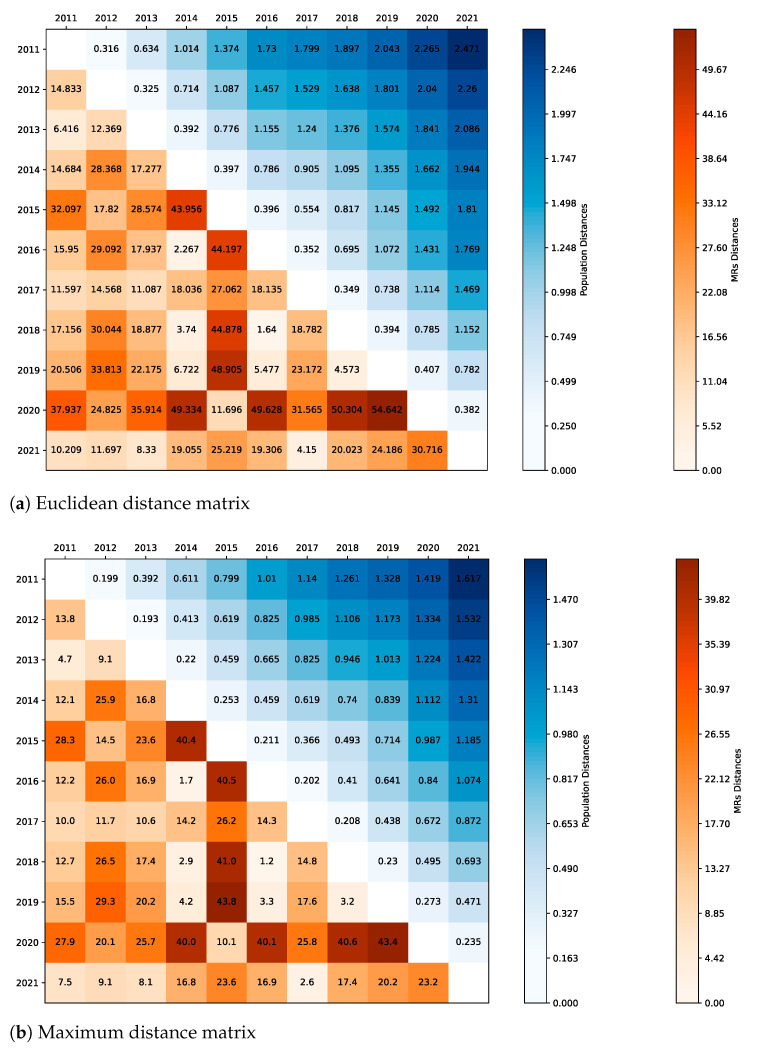
Adulthood distribution distance matrices related to the Euclidean (**a**) and maximum (**b**) distances. Authors’ elaboration on ISTAT mortality tables [11].

**Figure 5 ijerph-20-06481-f005:**
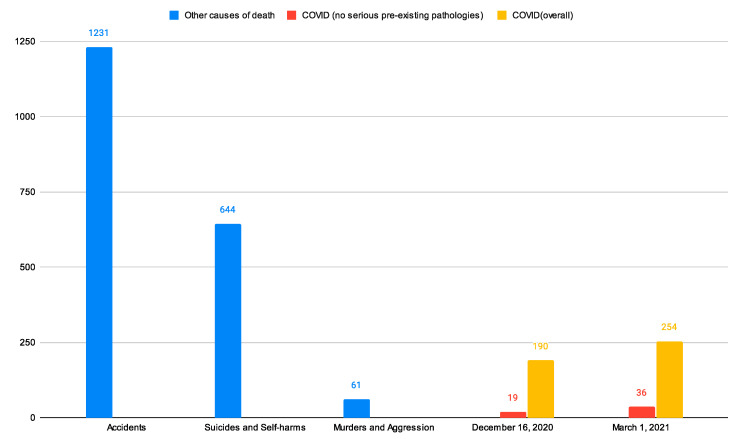
Comparison between the 2020 deaths in Italy for people aged 0–39 years old due to *external causes* and the number of deaths due to COVID-19 in 10 months (16 December 2020) and one year (1 March 2021), respectively. Authors’ elaboration on ISS reports [17,18] and ISTAT tables related to the causes of deaths [12].

**Figure 6 ijerph-20-06481-f006:**
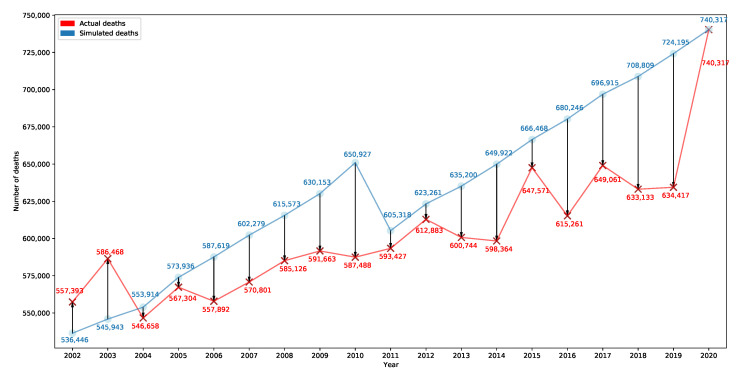
Chart associated with the answer to Question1. Authors’ elaboration on ISTAT databases related to the resident population and number of deaths [20,21,22,23]).

**Figure 7 ijerph-20-06481-f007:**
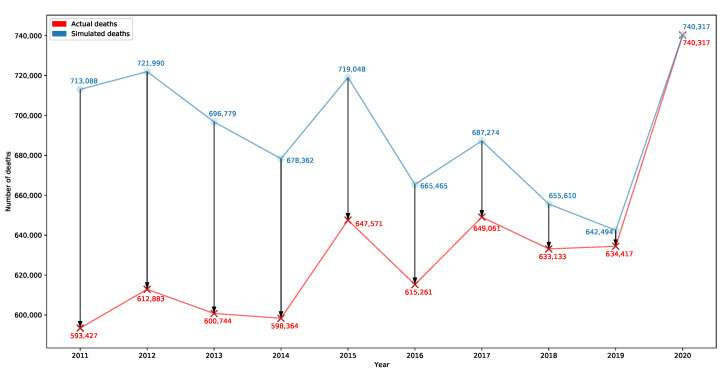
Chart associated with the answer to Question2. Authors’ elaboration on ISTAT databases related to the resident population and number of deaths [20,21,22,23]).

**Table 1 ijerph-20-06481-t001:** Total mortality rate (per thousand) per year reported by ISTAT [11].

Mortality Rate										
**2011**	**2012**	**2013**	**2014**	**2015**	**2016**	**2017**	**2018**	**2019**	**2020**	**2021**
9.9	10.2	10	9.9	10.8	10.2	10.8	10.6	10.6	12.5	11.9

**Table 2 ijerph-20-06481-t002:** Summary statistics on the MRs (per thousand) for the 50+ year old age groups in 2011–2019 with respect to 2020; μ and σ indicate the mean and the standard deviation, respectively. The 2020 MRs associated with a Z-score greater than |2| are highlighted in orange, and those associated with Z-scores greater than |3| in red. Authors’ elaboration on ISTAT mortality tables [11].

	Mortality Rate					
**Age** **Group**	**11–19** μ±σ	**11–19** Median	**11–19** Range	**11–19** IQR-Method	**2020** Z-Score	**2020**
50–54	2.5±0.1	2.5	(2.3, 2.7)	(2.1, 2.9)	0.0	2.5
55–59	4±0.2	4	(3.7, 4.3)	(3.6, 4.4)	0.0	4
60–64	6.4±0.3	6.4	(5.9, 6.8)	(5.35, 7.35)	0.94	6.7
65–69	10.2±0.5	10.4	(9.4, 10.9)	(9.4, 11)	1.43	10.9
70–74	16.6±0.7	17	(15.4, 17.4)	(14.95, 18.55)	2.41	18.3
75–79	28.8±1.4	19.3	(27.2, 31)	(24.65, 32.25)	2.31	32.1
80–84	54.4±3.0	54	(49.8, 58.4)	(45.55, 63.55)	1.31	58.3
85–89	104.2±4.3	104.9	(98.8, 110.2)	(88.7, 119.9)	2.18	113.5
90–94	187.6±5.9	184.8	(181.8, 199.6)	(176.1, 197.7)	3.75	209.7
95+	329.8±14.6	330.2	(314.7, 358.5)	(292.65, 360.25)	1.93	358.1

**Table 3 ijerph-20-06481-t003:** Ratios between the number of the 2020 non-COVID-19 first causes of death and the *COVID-19 first causes of death* associated with the Italian population aged 0–39 years old; NPP stands for No Pre-existing Pathologies, while O stands for Overall. Authors’ elaboration on ISS reports [17,18] and ISTAT tables related to the causes of deaths [12].

Non-COVID-19 Illness (Event)	COVID-19	
	* **O** *	* **NPP** *
External causes of injury and poisoning	7.67	54.14
Tumours	6.15	43.39
Accidents	4.85	34.19
Suicides and self-harms	2.54	17.89
Symptoms, signs, abnormal findings, and ill-defined causes	2.24	15.78
Congenital malformations and chromosomal abnormalities	1.54	10.86
Murders and aggression	0.24	1.69

**Table 4 ijerph-20-06481-t004:** Authors’ elaboration on ISTAT data [12] for two specific first causes of death: they are codified according to the ICD-10 guidelines reported by ISTAT [29]. The notation used here is the same as in Table 2.

	Number of Deaths				
**First Cause** **of Death**	**11–19** μ±σ	**11–19** Median	**11–19** Range	**11–19** IQR	**2020**
Symptoms, signs, abnormal results and ill-defined causes	12,490.4 ± 1793.3	12,541	(9984, 15,116)	(6413, 18,597)	24,709
Unknown and unspecified causes	2271 ± 375.9	2210	(1838, 2884)	(934, 3654)	9569

## Data Availability

Publicly available datasets were analyzed in this study. In particular, ISS and ISTAT datasets can be found at https://www.iss.it/ and https://www.istat.it/. More detailed and specific data sources are properly cited in this manuscript when needed.

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
