# Peer review of "Retrospective Analyses of COVID-19 and Population Ageing Effects on Italian Mortality during the Pandemic"

_ijerph, 2023, doi:10.3390/ijerph20156481_

Round 1

Reviewer 1 Report

Dear authors

Thank you for submitting your draft titled "Retrospective Analyses of COVID-19 and Population Ageing Effects on Italian Mortality During the Pandemic". I appreciate a lot your effort and  I am grateful to submit my comments and suggest changes to improve the paper.

My main concern about your work is the final general considerations, proposing in most cases, the communication methods used by scientists, journalists, and politicians turned out to be rather superficial and inaccurate,  making generalizations without having carried out an appropriate analysis, and planning adequate methodological tools been used for some of the conclusions he raises... I think that some paragraphs can be replanted and some lines and figures modified to consider the manuscript for publication.

I think the discussion and conclusion  needs focus only in your specific findings  and comment with detail without make generalizations based in some references without extensive review of this topic relate with public health interest issues communications by diferentt actors.

I consider the draft can be published making clarifications in the remarks that I add in the draft in the attached .pdf file.

Author Response

Dear reviewer,

we really appreciated all your suggestions. We attach here a '.pdf' file containing the answers to all your remarks present in 'peer-review-30820118.v1.pdf'.

Reviewer 2 Report

Dear Authors,

Thank you very much for your well-written manuscript, dealing with an important issue, which is the impact of the COVID-19 pandemic on the mortality of the Italian population in different age groups. Please find my comments and questions, pertaining to your manuscript:

1.      Lines 117-119: according to this data, no significant increase of mortality in the age groups 50-54 y.o. and 55-59 y.o. was observed during the pandemic. This makes difficult to understand your choice to present your data in younghood (0-49 y.o.) and adulthood (50+ y.o.) and not in younghood (0-59 y.o.) and adulthood (60+ y.o.). Please explain and discuss.

2.      Lines 130-131: how do you come to this conclusion so early in your analysis? I find that the provided information so far has not been enough to support this statement.

3.      Table 2 and Lines 442-444: how can you explain the non-significant increase of mortality in the age groups 80-84 and 95+ y.o. people? Is there a matter of distribution because of the division of the population in 5-years-groups or could you identify any particular reasons for that phenomenon?

4.      Lines 224-226: please explain your choice to examine the impact of the COVID-19 pandemic in younghood with and without pre-existing disease (COVID-19 Overall) and without pre-existing disease (COVID-19 NPP). It would have been much more reasonable to investigate the impact of the pandemic in younghood with serious pre-existing disease versus without pre-existing disease (COVID-19 NPP). Please explain and discuss.

Best Regards

Author Response

Dear reviewer,

Thanks for your valuable comments. Our manuscript's updated version based on your suggestion has been uploaded. In any case, we will answer all your requests point by point by mentioning the corresponding lines in the manuscript (where needed):

  1. The choice of splitting the population into 0-49 y.o. and 50-95+ is due to the fact that, in Italy, people aged 50+ y.o. were considered the critical target during the pandemic. For example, all the main COVID-19 prevention actions were devoted to this group (see, for example, the subsequent vaccination obligations). To clarify this point better, in lines 120-121, we added the following sentence:

               “””

               Nevertheless, we included the people aged 50+ y.o. in the adulthood                     category as it is well-known that this age group was considered the                       most critical during the pandemic.

               “””

  1. In lines 132-133, we actually meant to give a suggestion, i.e., a possible investigation direction which could seem to be the best to investigate. In any case, we modified the sentence:

              “””

              Thus, a significant and negative impact (i.e., more deaths) of COVID-19                  on mortality could be related only to 80+ years old age groups.

               “””

            with the following one:

               “””

               Thus, a significant and negative impact (i.e., more deaths) of COVID-19                 on mortality could be possibly related only to 80+ years old age                               groups.

               “””  

            Where the term “possibly” can help the reader to understand that 80+                  years old age groups could be (but not necessarily are) the most hit by                  COVID-19.

  1. Unfortunately, we cannot infer anything about this specific case, as we only limit ourselves to observing the available data: we read the data according to a neutral fashion. The 5-years-group was also used to be consistent with the official data. However, a possible explanation could rely on the fact that after exceeding the average life expectancy (in Italy, it is around 82 y.o.), i.e., a sort of critical threshold for many people, elders tend to stabilize their health situation facing new more quiet old age living conditions. 

  1. We have chosen to ‘examine the impact of the COVID-19 pandemic in younghood with and without pre-existing disease (COVID-19 Overall) and without pre-existing disease (COVID-19 NPP)’ as we wanted to be able to perform our analysis by looking at the ‘upper bounds’ of the number of COVID-19 deaths. In the presence of a diagnosis, one who suffers from a disease knows to belong to a group with pathologies. However, in the absence of a diagnosis, one cannot be excluded from suffering from a disease. Hence, the COVID-19 Overall group seems to us the best category to take into account even in the absence of information.              We summarized this motivation in our updated manuscript version in lines 233-234:

              “””

              Therefore, the COVID-19 O category allows us to look at the worst                        possible case, even without specific patient information. 

              “””

           Furthermore, in our work, we specified that the ”COVID-19 Overall”                       category also includes patients positive “with no clinic info available”                     (lines 231-232).

Round 2

Reviewer 2 Report

Dear Authors,

thank you for providing comprehensive and convincing answers to my questions and queries and made changes, which have contributed to the optimization of your manuscript and increased the publishing potential of your work.

Best Regards